

# A novel baculovirus-derived promoter with high activity in the baculovirus expression system

María Martínez-Solís[1,2], Silvia Gómez-Sebastián[3], José M. Escribano[4], Agata K. Jakubowska[1] and Salvador Herrero[1,2]

[1] Department of Genetics, Universitat de València, Burjassot, Spain
[2] Estructura de Recerca Interdisciplinar en Biotecnologia i Biomedicina (ERI BIOTECMED), Universitat de València, Burjassot, Valencia, Spain
[3] Alternative Gene Expression S.L. (ALGENEX), Madrid, Spain
[4] Departamento de Biotecnología, Instituto Nacional de Investigación y Tecnología Agraria y Alimentaria (INIA), Madrid, Spain

Corresponding author
Salvador Herrero, sherrero@uv.es

## ABSTRACT

The baculovirus expression vector system (BEVS) has been widely used to produce a large number of recombinant proteins, and is becoming one of the most powerful, robust, and cost-effective systems for the production of eukaryotic proteins. Nevertheless, as in any other protein expression system, it is important to improve the production capabilities of this vector. The *orf46* viral gene was identified among the most highly abundant sequences in the transcriptome of *Spodoptera exigua* larvae infected with its native baculovirus, the *S. exigua* multiple nucleopolyhedrovirus (SeMNPV). Different sequences upstream of the *orf46* gene were cloned, and their promoter activities were tested by the expression of the GFP reporter gene using the *Autographa californica* nucleopolyhedrovirus (AcMNPV) vector system in different insect cell lines (Sf21, Se301, and Hi5) and in larvae from *S. exigua* and *Trichoplusia ni*. The strongest promoter activity was defined by a 120 nt sequence upstream of the ATG start codon for the *orf46* gene. On average, GFP expression under this new promoter was more than two fold higher than the expression obtained with the standard polyhedrin (polh) promoter. Additionally, the *orf46* promoter was also tested in combination with the polh promoter, revealing an additive effect over the polh promoter activity. In conclusion, this new characterized promoter represents an excellent alternative to the most commonly used baculovirus promoters for the efficient expression of recombinant proteins using the BEVS.

## INTRODUCTION

Baculoviruses are enveloped, double-stranded DNA viruses pathogenic to invertebrates, preferably Lepidoptera. Their specificity to kill a narrow spectrum of insects and their safety for humans, plants, and non-target insects, make them a good biological control agent. In addition, since 1983, baculoviruses have been extensively used as protein expression vectors in insect cells (*Smith, Summers & Fraser*, *1983*). The baculovirus expression vector

system (BEVS) has been widely used to produce a large number of recombinant proteins, and several systems using different strategies for the generation of the recombinant viruses have been developed (*Li et al.*, *2012*; *Van Oers, Pijlman & Vlak*, *2015*). The high popularity reached by this system is due to its ability to produce large amounts of active proteins, together with its ability to introduce post-translational modifications in the expressed protein, similar to mammalian cells, such as glycosylation or phosphorylation (*O'Reilly, Miller & Luckow*, *1994*).

Similar to most viruses, the baculovirus gene expression has a temporal regulation which can be divided into three main phases: the early, late, and very late phases (*Friesen*, *1997*; *Lu & Miller*, *1997*; *Jarvis*, *2009*). The expression of the early genes does not require prior viral protein synthesis and precedes viral DNA replication. The late phase is a period for viral DNA replication, and the very late phase is characterized by the production of viral particles. In this final phase of infection the expression of the polyhedrin and p10 structural proteins predominate, and these comprise the major proportion of the cell protein mass. The high transcription yield of the promoters of these two proteins has been exploited in the BEVS to express foreign proteins (*Rohrmann*, *1999*). The baculovirus of *Autographa californica* (*A. californica* nucleopolyhedrovirus, AcMNPV) is the main viral species used as an expression vector for recombinant protein expression using the BEVS. The polyhedrin and the p10 promoters from AcMNPV have been extensively used for the expression of foreign proteins with this system. However, recombinant protein expression yields not only depend on the promoter used, but also on the host cell line, as well as the characteristics of the foreign gene (*Morris & Miller*, *1992*). Several strategies have been developed to improve the production of functional proteins in insect cells. For instance, modification of the expression vectors by the addition of DNA elements involved in protein expression processes can enhance the production yields of recombinant proteins (*Lo et al.*, *2002*; *Venkaiah et al.*, *2004*; *Manohar et al.*, *2010*; *Tiwari et al.*, *2010*; *Gómez-Sebastián, López-Vidal & Escribano*, *2014*). Nevertheless, one of the main cis-regulatory elements affecting the protein expression levels is the promoter. To date, different types of promoters have been tested in the BEVS to improve recombinant protein expression. Viral promoters such as vp39 or 39K, and promoters derived from insect larvae such as the hexamerin-derived promoter pB2 from *Trichoplusia ni* (*López-Vidal et al.*, *2013*) showed high levels of expression of recombinant proteins. In other cases, the combination of some of these promoters with the conventional promoters exhibited higher expression levels of the recombinant proteins than the standard late promoters alone (*Thiem & Miller*, *1990*; *Morris & Miller*, *1992*; *Ishiyama & Ikeda*, *2010*; *Lin & Jarvis*, *2012*).

In a previous work, the transcriptional pattern of the *Spodoptera exigua* multiple nucleopolyhedrovirus (SeMNPV) during the infective process in its natural host revealed very high levels of expression for the *orf46* viral gene (*Pascual et al.*, *2012*). Since the *orf46* gene codes for the structural protein polyhedron envelope protein (PEP), we hypothesized that its expression could be regulated by a strong promoter. In this study, we have determined the core regulatory sequence for the gene (*orf46*) derived from the SeMNPV and we have examined its ability to drive the expression of recombinant proteins in insect cells using the BEVS. Different sequences upstream of the ATG start codon of the *orf46* gene were cloned,

and their promoter activities were tested by the expression of GFP as a reporter gene using the AcMNPV system in different insect cell lines. In addition, the promoter activity of this region was tested when combined with the standard polyhedrin promoter derived from the AcMNPV.

## MATERIALS AND METHODS

### Culture cells and insects

The *Spodoptera exigua* (Se301) and *Spodoptera frugiperda* (Sf21) cell lines were cultured at 25 °C in Gibco® Grace's Medium (1X) (Life technologies™) supplemented with 10% heat-inactivated fetal bovine serum (FBS). The *Trichoplusia ni* (High Five, Hi5) cell line was cultured at 27 °C in TNMFH medium supplemented with 10% FBS and gentamicin (50 μg/ml). *S. exigua* larvae were maintained in the laboratory, reared on an artificial diet at 25 ± 3 °C with 70 ± 5% relative humidity and a photoperiod of 16/8 h (light/dark). *Trichoplusia ni* (cabbage looper) larvae were reared on an artificial insect diet and were kept in growth chambers at 22 ± 1 °C under controlled humidity (50%) and light period (8 h/day) conditions.

### Sequence identification

The transcriptional regulatory region was determined by *in silico* analysis of the sequences derived from the Roche 454 FLX and Sanger methods obtained from the transcriptome of *S. exigua* larvae which included samples of SeMNPV-infected larvae (at the latest stage of the infection) (*Pascual et al.*, *2012*). First, the ten ORFs with the highest expression levels were obtained based on their maximum coverage. Then, the upstream region from the ATG start codon of *orf46* was analyzed *in silico* and manually for the prediction of the transcriptional regulatory region. Using promoter prediction software (http://www.fruitfly.org/seq_tools/promoter.html), we identified a transcription start site and other motifs characteristic for baculovirus promoters. A sequence of 300 bp upstream of the predicted start codon was selected as an initial candidate region to act as a promoter.

### Construction of recombinant baculoviruses

Several baculovirus-transfer plasmids containing different fragments of the 5′ region of the *orf46* gene driving the expression of GFP were generated using the AcMNPV vector system (Fig. 1). The GFP gene was initially cloned under the control of the polyhedrin promoter (polh) to generate the pFB-PL-GFP vector (*López-Vidal et al.*, *2013*) (from now, polh-GFP). The initial pSeL and pSeS promoter sequences were obtained by PCR amplification using SeMNPV genomic DNA as template. PCR amplifications were performed using specific primers which added *Bstz17*I and *Spe*I restriction sites. The polh promoter was then replaced by the pSeL or pSeS fragments into the *Bstz17*I and *Spe*I sites, to generate the pSeL-GFP and pSeS-GFP vectors, respectively. The pSeL140 and pSeL120 sequences were amplified by PCR from the pSeL-GFP vector using two specific primers. The first primer included the corresponding 5′ region of pSeL and a *BstZ17*I restriction site, and the second primer was designed to amplify from a 3′ region of the GFP gene containing an *Avr*II restriction site. These sequences were cloned into the *BstZ17*I and *Avr*II sites of the polh-GFP vector by replacement of the polh promoter, generating the pSeL140-GFP and pSeL120-GFP vectors.

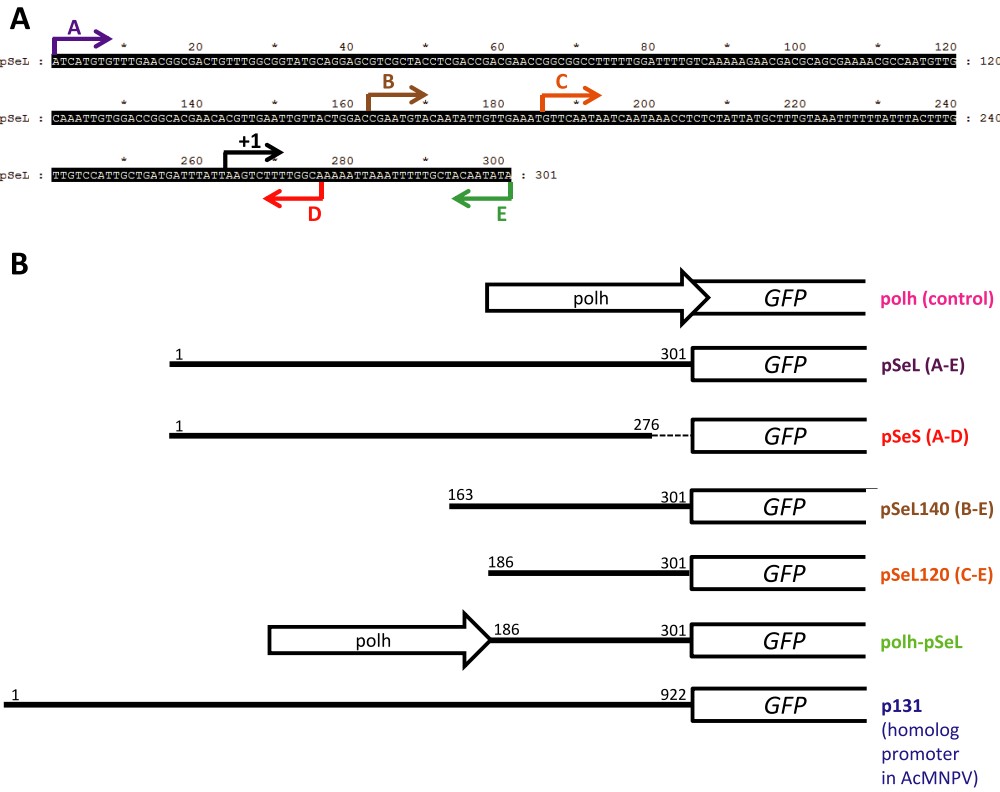

**Figure 1** **Schematic summary of the recombinant baculoviruses carrying different promoter regions employed in this study.** (A) Nucleotide sequence upstream of the *orf46* gene from SeMNPV selected as a regulatory region (nucleotides 45417–47500 from AF169823). The arrows indicate the range of the fragments from the 5′ to 3′ sequence that were tested for promoter activity and the transcription initiation site (+1). (B) Schematic representation of the recombinant baculoviruses generated which carry different fragments of the sequence upstream of the *orf46* gene to test their promoter activity using GFP as a reporter. The white open arrows (Polh) represent the polyhedrin promoter. The white boxes represent the GFP gene. The numbers indicate the first and the last nucleotides (from 5′ to 3′) of the sequence that was cloned as a promoter. The dotted line in the pSeS construct represents the 5′ fragment that is absent.

The vector combining two promoters (polh-pSeL-GFP) was constructed by modification of the polh-GFP vector. The pSeL120 promoter fragment was obtained by PCR using specific primers which added *Xho*I and *Avr*II restriction sites. The resulting fragment was inserted into the *Xho*I and *Avr*II sites of the polh-GFP vector, generating the polh-pSeL-GFP vector containing both of the polh and pSeL120 promoters in tandem. Additionally, the DNA sequence corresponding to the p131 (homolog to *orf46* in AcMNPV) promoter was chemically synthesized (GenScript) and flanked by *BstZ17*I and *Spe*I restriction sites. This was cloned into a pFB vector to control the expression of the GFP gene, generating the p131-GFP vector. Figure 1 shows a schematic representation of all of the different recombinant baculoviruses generated in the present work. The sequences of the primers employed for the cloning of the different constructs are summarized at Table S1.

The recombinant baculoviruses were obtained using the Bac-To-Bac® baculovirus expression system (Invitrogen, Carlsbad, CA, USA) following the manufacturer's instructions. Plasmids generated in the previous step were used to transform *E. coli*

DH10Bac$^{TM}$ heat-shock competent cells and generate the corresponding recombinant bacmids. Bacmids were purified and used to transfect Sf21 cells using Cellfectin® II Reagent (Invitrogen) following the manufacturer's instructions. The resulting baculoviruses were collected after 4–5 days of incubation at 27 °C. These baculoviruses were amplified once to obtain high-titer stocks for further experiments, and the viral titers were determined by quantitative PCR (qPCR) using specific primers (Table S1). For that purpose, viral DNAs were treated using Prepman reagent (Applied Biosystems) following the manufacturer's instructions and were quantified by comparing the obtained Ct values against a standard curve of known viral concentration. Viral titers used in the standard curve were obtained by end point dilution, a method that does not consider non-infective viruses. The viral titers were expressed as baculoviruses per milliliter (BVs/ml).

## Infection assays in culture cells and insects

Cells (Se301, Sf21, and Hi5) were cultured in 24-well plates at a confluence of 70%, then the cells were infected with the different recombinant baculoviruses at a multiplicity of infection (MOI) of 5. The cells were collected at different times post-infection by low speed centrifugation (3,000 rpm, 5 min) to avoid cell lysis, and kept at –20 °C until the quantification of GFP expression. Last instar *S. exigua* and *T. ni* larvae were injected with 5 µl of recombinant baculoviruses containing $5 \times 10^4$ BVs. Larvae were maintained at 25 °C and 28 ° C, respectively, and after 72 h post-infection (hpi) were frozen at –20 °C until they were processed for GFP quantification.

## Analysis of GFP expression

Frozen cells from the infection assays were resuspended in a lysis buffer (50 mM Tris–HCl pH 7.5, 100 mM NaCl, 1 mM DTT, 5% glycerol), incubated for 5 min at room temperature, and centrifugated at $16,000 \times g$ for 1 min. The supernatant was collected to measure GFP expression by fluorescence in a microplate reader (Infinite® 200 PRO NanoQuant; TECAN, Männedorf, Switzerland) (excitation 485 nm, emission 535 nm). Each value was obtained by measuring each sample 4 times. Occasionally, the production of GFP was confirmed by direct observation of the GFP protein band in SDS-PAGE, suggesting a good correlation between GFP intensity and protein abundance. Frozen larvae were homogenized in 1 ml of extraction buffer (0.01% de Triton X-100, 1 mM de PMSF, and DTT 25 mM in PBS $1\times$). Homogenates were centrifuged at $1,800 \times g$ for 30 min at 4 °C, and the supernatant was collected to measure GFP as described above. The values correspond to at least two independent replicates for all of the experiments. Statistical analyses were performed by Dunnett's Multiple Comparison Test using the GraphPad Prism program (GraphPad software Inc., San Diego, CA, USA).

# RESULTS

## Expression of viral genes and promoter selection

Expression levels of the SeMNPV genes were monitored by mapping of the viral reads on the transcriptome of *S. exigua* infected larvae (Pascual et al., 2012). As expected, the most abundant reads were mapping on the *orf1* which corresponded to the polyhedrin gene
**Table 1** ORFs from the SeMNPV highly expressed during infection of *S. exigua* larvae (*Pascual et al.*, *2012*).

| ORF | Description | Coverage[a] |
| --- | --- | --- |
| ORF1 | Polyhedrin | 674 |
| ORF46 | Calyx/polyhedron envelope protein | 590 |
| ORF127 | lef6 | 516 |
| ORF122 | – | 416 |
| ORF94 | – | 347 |
| ORF71 | odv-e25 | 344 |
| ORF65 | p6.9 DNA binding protein | 262 |
| ORF136 | odv-e18 | 255 |
| ORF32 | pkip | 224 |
| ORF124 | – | 204 |

**Notes.**

[a] Coverage reported as the maximum coverage (number of reads) for a given ORF after mapping of the SeMNPV genome with transcriptional data.

(Table 1). The second most abundant ORF mapped, corresponded to the *orf46* gene of SeMNPV. *Orf46* codes for the polyhedron envelope protein (PEP), a structural protein that surrounds the polyhedra of the viral particles. In addition, other genes highly expressed during the infection were *orf127* and *orf122*. Given the high expression observed for the *orf46* gene under our experimental conditions, and its role as a structural protein, we decided to explore the possibility of using its regulatory sequence as a promoter for foreign gene expression using the BEVS.

Detailed analysis of the 454-derived reads mapping to the SeMNPV genome predicted the transcription start site (site +1) of *orf46* at position 89150 (which referred to the reverse complementary SeMNPV genome, GenBank acc: AF169823.1). A region of 301 nt upstream of the start codon of *orf46* from SeMNPV was initially selected as the promoter sequence. The *in silico* analysis predicted a promoter between nucleotides 224–269 from the selected sequence that revealed the presence of a TAAG motif. This TAAG motif was in an AT rich region, and it was described as a typical transcriptional initiation site of late and very late baculovirus promoters (*Lu & Miller*, *1997*).

## Orf46 promoter activity in insect cells

To determine the promoter activity of the region upstream of the *orf46* gene from SeMNPV and its homologous equivalent region in AcMNPV (p131), different constructs were obtained and tested for their ability to drive the expression of the GFP reporter gene (Figs. 2 and 3) using the BEVS. Se301, Sf21, and Hi5 cells were infected with recombinant AcMNPV baculoviruses expressing GFP under the different promoter regions, and their activities were compared to the activity obtained with the standard polyhedrin (polh) promoter from AcMNPV. The GFP expression yields obtained for the 300 bp fragment upstream of *orf46* (pSeL) was equivalent to that obtained with the polh promoter in the Se301 and Sf21 cells. Interestingly, for the Hi5 cells, the GFP expression mediated by the pSeL promoter was about two-fold higher than that obtained using the polh promoter. The deletion of the

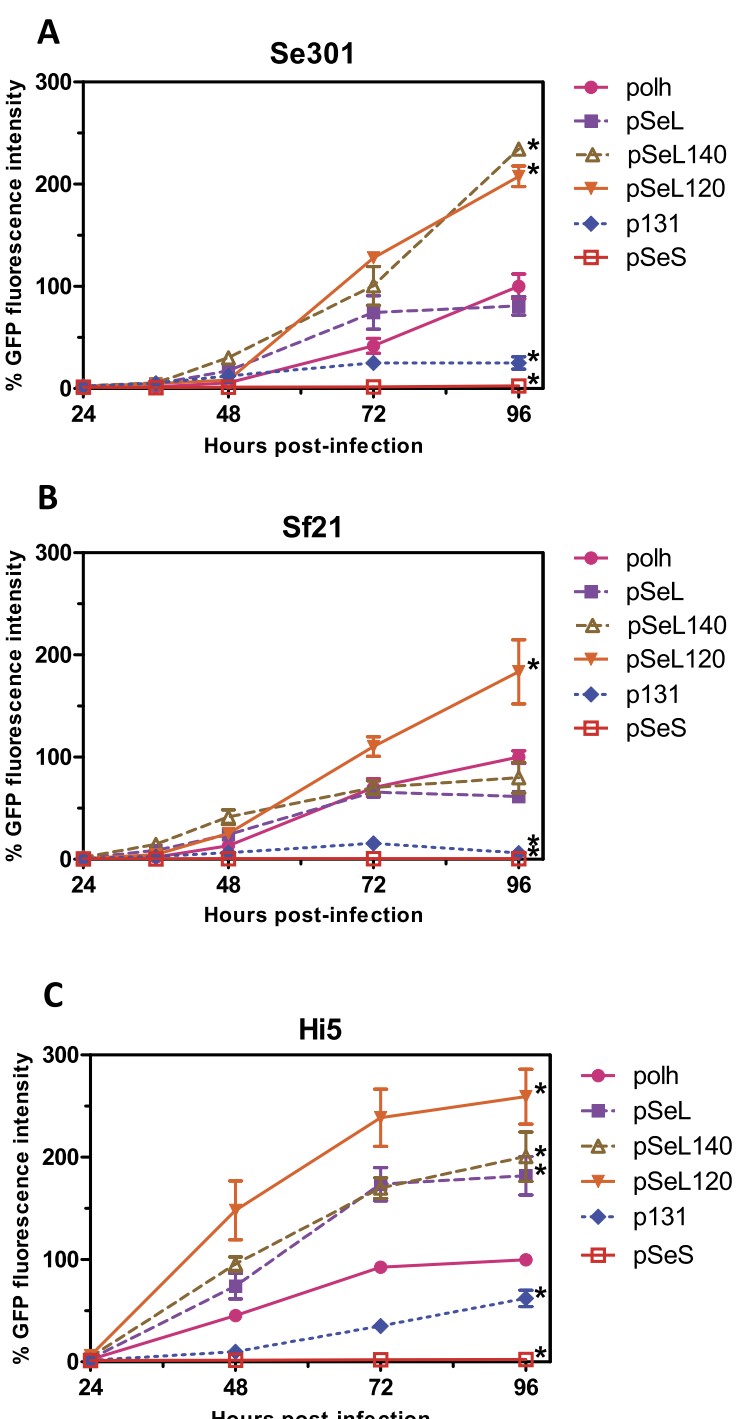

**Figure 2** **Promoter activity of the sequences upstream of the *orf46* gene.** GFP expression, measured as relative fluorescence intensity, in different insect cell lines infected with the different baculoviruses at a multiplicity of infection (MOI) of 5. The fluorescence was measured at different time points after infection of Se301 (A), Sf21 (B), and Hi5 (C) cells. The results are expressed as the relative percentage of GFP fluorescence intensity, taken as 100% of the value corresponding to the maximum intensity obtained with the polh promoter. The values are the means of at least two independent assays. The error bars represent the standard error of the mean.

| pSeS | polh | pSeL120 | polh-pSeL |
|---|---|---|---|

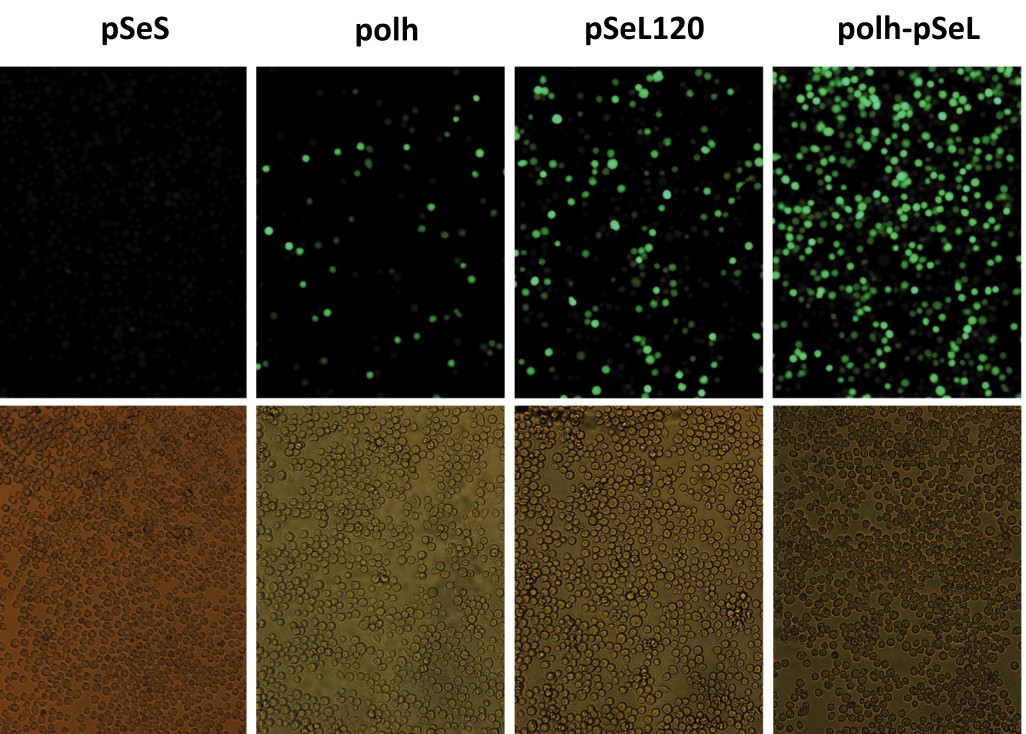

**Figure 3** **Fluorescence microscopy of Sf21 cells infected with the different baculoviruses.** A representative image of Sf21 cells infected with a selected baculovirus at a MOI of 5. The images were taken at 48 h post-infection.

25 nucleotides in the 3′ region of the pSeL sequence (pSeS) strongly affected its promoter activity in the three cell types tested, revealing the importance of this region in the activity of the *orf46* promoter. The homolog promoter in AcMNPV (p131) showed a significantly lower expression level than pSeL and the control polh in all of the cell lines tested (Fig. 2).

In order to further delimit the promoter region, two additional constructs containing 120 and 140 nucleotides upstream of the *orf46* gene were also tested (pSeL120 and pSeL140, respectively). The reduction in the promoter size had a positive impact on the promoter activity in most cases, as the GFP expression was double that seen when compared to the polh promoter (Figs. 2 and 3). The highest expression levels were observed for the region consisting of the 120 nt upstream of the *orf46* gene. When compared to the polh promoter, pSeL120 showed an increase in expression of more than two fold in all of the cell lines tested. These results strongly suggest that pSeL120 could be considered a useful promoter with the capacity to significantly increase the expression yields obtained with the conventional polyhedrin promoter in the BEVS.

## Activity of the pSeL120 in combination with standard promoter in insect cells

In a subsequent analysis, a recombinant baculovirus expressing GFP under the control of a promoter combining the pSeL120 and polh in tandem (polh-pSeL) was generated and tested for its expression levels. After infecting insect cells with this recombinant baculovirus, we

observed an additive effect over the two promoters used separately in different recombinant baculoviruses, increasing the polh-pSeL promoter GFP expression to around 2-fold of the levels obtained with the polh promoter alone (Figs. 3 and 4). This additive effect was observed with small variations in the three insect cell lines tested. These results revealed the potential of pSeL120 to be combined with other promoters in order to produce increased amounts of recombinant proteins in the BEVS.

## Activity of the new promoters in baculovirus-infected insect larvae

Although BEVS is mainly used for protein production in insect cell cultures, they can also be used to efficiently produce recombinant proteins in a cost-effective manner by using Lepidoptera larvae. We tested the activity of several of the above described new promoters in larvae from two species of Lepidoptera, the specific host of the SeMNPV, *S. exigua* and *T. ni*, commonly used for protein production using AcMNPV-based vectors. The last instar of *S. exigua* and *T. ni* larvae were infected by intrahemocelical injection with the recombinant baculoviruses expressing GFP under the control of every promoter tested. After 48 hpi, the protein production was estimated by measuring the GFP fluorescence of the larval extracts (Fig. 5). For all of the viruses tested in both insect species, the GFP production using pSeL or pSeL120 was equivalent to that obtained with a baculovirus expressing this protein under the control of the polyhedrin promoter. For the baculovirus comprised of both the polh and pSeL120 promoters, the expression was similar to the polh control promoter in *T. ni* larvae, and slightly lower in *S. exigua* larvae.

## DISCUSSION

Despite the wide use of the BEVS since the early 1980s (*Smith, Summers & Fraser*, *1983*), the system remains in terms of productivity very similar to the one originally developed. It is worth pointing out the need for research in the improvement of the productivity by different approaches, as was previously shown for other eukaryotic and prokaryotic production platforms. Several strategies have been attempted to increase the production yields by introducing modifications and improvements at different levels. Some of the improvements in the BEVS have been focused on the modification of viral promoters (*Manohar et al.*, *2010*), or the introduction of regulatory sequences (*Sano et al.*, *2002*; *Tiwari et al.*, *2010*; *Ge et al.*, *2014*; *Gómez-Sebastián, López-Vidal & Escribano*, *2014*). Other strategies were based on the deletion of non-essential genes of the vector (*Hitchman et al.*, *2010*; *Hitchman et al.*, *2011*). One standard strategy is the search for promoters which are stronger than those commonly used, such as the p10 and polyhedrin (polh) promoters, or chimeras of them employed in laboratory and industrial production (*Thiem & Miller*, *1990*; *Ishiyama & Ikeda*, *2010*; *Lin & Jarvis*, *2012*; *López-Vidal et al.*, *2013*). However, often the efficiency of the promoter also depends on the regulatory sequences around them and the type of cellular lines in which they are acting (*Matsuura et al.*, *1987*; *Morris & Miller*, *1992*; *Gross & Rohrmann*, *1993*; *Lo et al.*, *2002*). Thus, the development of new promoters to implement the cost-efficient production of recombinant proteins and to provide alternatives to the traditional promoters, still remains of interest.

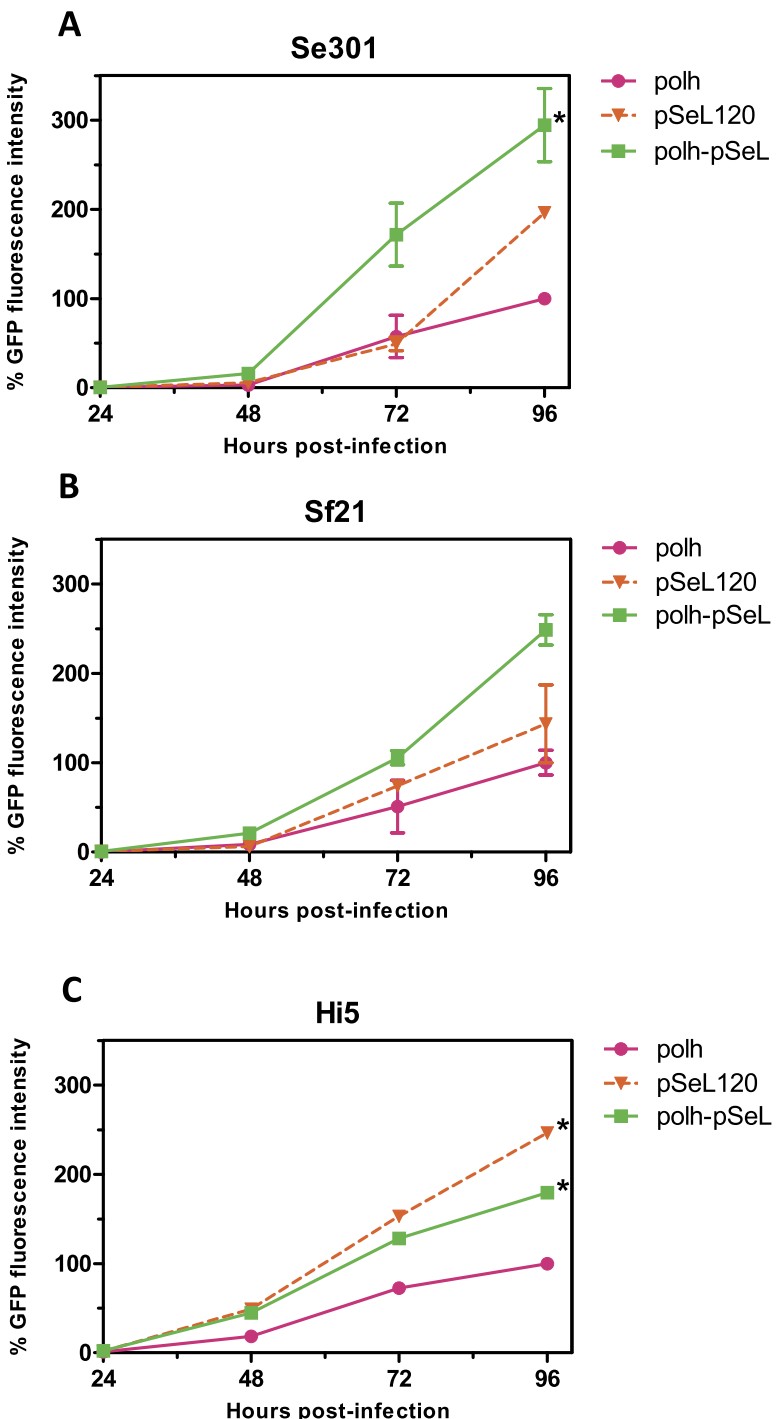

**Figure 4  Promoter activity of pSeL120 when combined with the pph promoter.** GFP expression, measured as the relative fluorescence intensity, in different insect cell lines infected with the different baculoviruses at a MOI of 5. The fluorescence was measured at different time points after the infection of Se301 (A), Sf21 (B), and Hi5 (C) cells. The results are expressed as the relative percentage of GFP fluorescence intensity, taken as 100% of the value corresponding to the maximum intensity obtained with the polh promoter. The values are the means of at least two independent assays. The error bars represent the standard error of the mean.

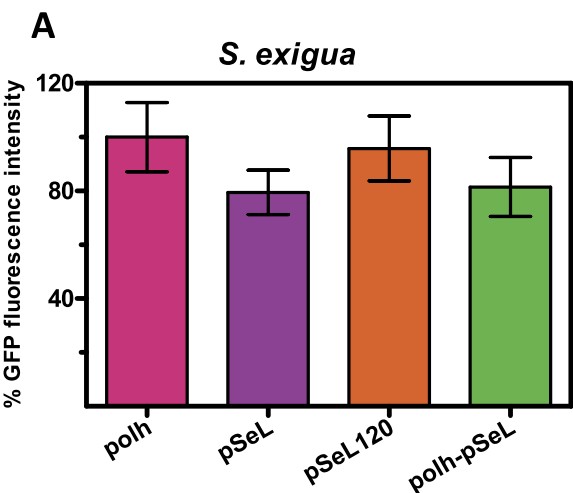

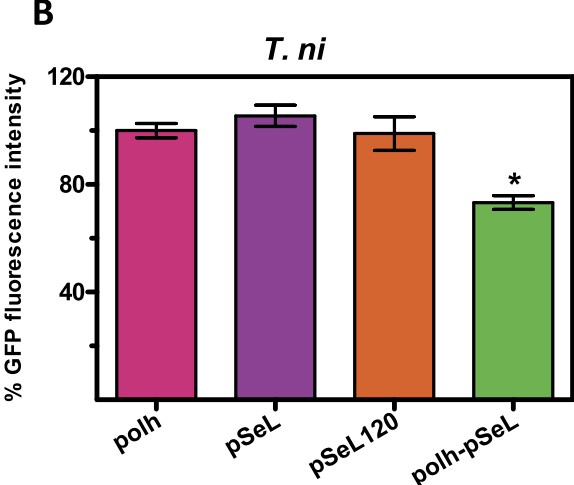

**Figure 5** **Promoter activity in insect larvae.** GFP expression, measured as relative fluorescence intensity, was obtained in insect larvae infected with the different recombinant baculoviruses. The results are expressed as the relative percentage of GFP fluorescence intensity, taken as 100% of the value corresponding to values for the control sequence with the polh promoter. The values are the means of at least two independent assays. The error bars represent the standard error of the mean.

Viral genes coding for structural proteins are usually regulated by strong promoters, since they need to be highly translated to produce the viral particles. Thus, they are good candidates to explore in the improvement of the BEVS. The promoter studied in this work regulates the expression of the *orf46* gene from SeMNPV, which codes for the calyx/polyhedron envelope protein (PEP). The polyhedron envelope is an electron-dense structure that forms a smooth, seamless surface that surrounds polyhedra. The function of calyx/PE is to seal the surface of polyhedra and to enhance their stability (*Rohrmann*, *2013*). Homologs of the PEP are found in the genomes of all lepidopteran nucleopolyhedroviruses. The PEP is associated with p10 fibrillar structures, and both proteins appear to be important for the proper formation of

the polyhedron envelope (*Van Lent et al.*, *1990*; *Russell, Pearson & Rohrmann*, *1991*; *Gross, Russell & Rohrmann*, *1994*; *Lee et al.*, *1996*). PEP from AcMNPV was shown to be associated with BV but not with ODV. It is abundantly produced during the late phase of infection (*Wang et al.*, *2010*).

In the present study, we have described a new viral promoter sequence derived from the gene that codes for the structural PEP from SeMNPV, showing better performance than the polh promoter in the BEVS in different cell lines. By testing different sequences upstream of the ATG start codon from the *orf46* gene driving the expression of GFP, we have limited the essential promoter sequence. The sequence corresponding to the 120 nt just before the ATG start codon (pSeL120) showed the strongest promoter activity when it was functioning in cultured cell lines. On average, the expression under the pSeL120 promoter was at least 2 times higher than the maximum expression levels reached using the standard polh promoter. Other groups have investigated the characterization of new promoters for increased expression yields. *Lin & Jarvis* (*2012*) showed that the delayed early 39K promoter from AcMNPV produced 4-fold more SEAP protein than the polyhedrin promoter in Sf21 cells. *López-Vidal et al.* (*2013*) isolated the pB2 promoter (promoter region of the Basic juvenile hormone-suppressible protein 2, BJHSP-2) from the Lepidoptera *T. ni* with activity in Sf21 cells. The pB2 promoter can drive the expression of GFP earlier in time, but it is not as strong as the polyhedrin promoter. *Ishiyama & Ikeda* (*2010*) reported that the expression of GFP was increased using the vp39 late promoter in comparison to the polyhedrin promoter in *Bombix mori* cultured cells.

Despite the high level of conservation and similarity between sequences from different virus species, the homolog p131 sequence from AcMNPV showed the lowest promoter activity, even lower than the control polh promoter. Such discrepancy could be explained by the fact that the p131 transcription start site (predicted *in silico)* is not located in the TAAG region, and this region seems to be very important in order to obtain high expression levels (as mentioned above). Alternatively, it could also be possible that the activity of p131 in AcMNPV is not as crucial as the orf46 activity in SeMNPV. This hypothesis is supported by some gene expression data in AcMNPV in the literature. It has been published that the gene expression levels of pp34 (gene whose expression is controlled by p131 in AcMNPV) were considerably lower than the polyhedrin and p10 expression in infected Sf9 cells (*Iwanaga et al.*, *2004*). The analysis of the transcriptome of AcMNPV-infected *T. ni* cells also showed lower expression levels of pp34 in comparison to the polyhedrin and p10 genes (*Cheny et al.*, *2013*).

An additional improvement with regard to the protein expression was obtained when the pSeL120 promoter was combined with the polh promoter (polh-pSeL), resulting in increases of about 2-fold over the polh promoter and 1.5 fold over the pSeL120 promoter alone. Increases in protein production have also been reported by the combination of different promoters. *Thiem & Miller* (*1990*) showed that the combination of the vp39 and the polyhedrin promoter enhanced the expression of foreign genes compared to using those promoters alone in Sf cells, because this hybrid promoter showed regulation patterns of late and very late promoters. *López-Vidal et al.* (*2013*) also demonstrated an increase in GFP production of more than 20% at early times post-infection, and similar expression levels

at very late times post-infection in Sf21 cells using a pB2-p10 promoter combination, with respect to conventional late promoters.

Although our results have shown a clear improvement of the pSeL promoter activity in different cell types, we could not observe such improvement when it was used for protein production in *S. exigua* and *T. ni* larvae. The difference in the promoter activity between the cell lines and larvae could be due to additional factors affecting the replication dynamics and/or promoter activity of the virus, as well as the timing selected for the processing of the larvae. Nevertheless, the pSeL120 promoter activity in larvae is equivalent to that obtained using the polh promoter, and no significant differences were observed, demonstrating that the promoter exhibits versatility and can be utilized in both cell lines (with high activity for a wide range of cell types) and insect larvae (with activity equal to the polh promoter).

When compared with homologous sequences in other viral species, we found a region of 50 nt upstream of the ATG start codon that was highly conserved between them. Interestingly, removal of 25 nt of this sequence downstream of the +1 start transcription site in mRNA abolishes the activity of the pSeS promoter. This observation suggests that this region is essential for the strong promoter activity as already proposed in previous studies. *Weyer & Possee* (*1988*) showed that the 5′ UTR regions are necessary for the maximum activity of the polyhedrin and p10 promoters. In agreement with that, expression levels for foreign proteins are related to the integrity of the 5′ UTR region of the polyhedrin gene (*Matsuura et al.*, *1987*; *Luckow & Summers*, *1988*). The sequence located between the TAAG motif and the translation initiation site is known in baculoviruses as the burst sequence (BS) (*Weyer & Possee*, *1988*). This is a sequence of about 50 nt required for the efficient expression of viral genes during the very late phase of infection. Studies of mutational analysis regarding the BS region have demonstrated that BS are essential for efficient protein expression (*Ooi, Rankin & Miller*, *1989*; *Weyer & Possee*, *1988*), which agrees with our results. If we take into consideration that most of those highly conserved 50 nucleotides are included in the 5′ UTR of the ORF46 transcript, it seems that the increase in expression found with the pSeL-derived promoters is likely influenced by the effect of such sequences with respect to the access provided to the RNApol, which affects the transcription and translation rates, and even increases mRNA stability. The reason in this case is not known, but it has already been described that an upstream sequence of the AcMNPV polyhedrin gene has an important function for mRNA transcription and translation efficiencies (*Min & Bishop*, *1991*).

In conclusion, the sequence derived from the SeMNPV genome described in this work represents a new promoter which is able to express, in most cases, higher yields of foreign proteins than the polh promoter in the BEVS. Moreover, the combination of pSeL with the conventional polh promoter showed higher activity for the expression of GFP than the pSeL or polh promoters alone. Although additional validations of this promoter for the expression of recombinant proteins other than GFP would be needed, these results represent a new improvement in the production of recombinant proteins using the BEVS, with potential application in the cost-efficient large-scale industrial production of biologics.

# ACKNOWLEDGEMENTS

We want to thank Rosa Maria González-Martínez for her excellent help with insect rearing and laboratory management.

### Funding

This study received financial support from the Spanish Ministry for Science and Technology (AGL2011-30352-C02, AGL2013-48550-C2-2-R, and AGL2014-57752-C2). The funders had no role in study design, data collection and analysis, decision to publish, or preparation of the manuscript.

### Grant Disclosures

The following grant information was disclosed by the authors:
Spanish Ministry for Science and Technology: AGL2011-30352-C02, AGL2013-48550-C2-2-R, AGL2014-57752-C2.

### Competing Interests

The use of the promoter described in this manuscript has been patented (Patent No. ES 2554561 Spanish Office of Patents and Trademarks). In the course of the investigation, Siliva Goméz-Sebastián was an employee of Alternative Gene Expression S.L. (ALGENEX). The authors declare there are no competing interests.

### Author Contributions

- María Martínez-Solís conceived and designed the experiments, analyzed the data, wrote the paper, prepared figures and/or tables, reviewed drafts of the paper.
- Silvia Gómez-Sebastián and Agata K. Jakubowska performed the experiments, analyzed the data, reviewed drafts of the paper.
- José M. Escribano reviewed drafts of the paper.
- Salvador Herrero conceived and designed the experiments, performed the experiments, analyzed the data, wrote the paper, prepared figures and/or tables, reviewed drafts of the paper.

### Patent Disclosures

The following patent dependencies were disclosed by the authors:
Patent No. ES 2554561 Spanish Office of Patents and Trademarks.

### Data Availability

The raw data has been supplied as Supplemental Dataset.

### Supplemental Information

Supplemental information for this article can be found online at http://dx.doi.org/10.7717/peerj.2183#supplemental-information.

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
