# Peer review of "A novel baculovirus-derived promoter with high activity in the baculovirus expression system"

_PeerJ, doi:10.7717/peerj.2183_

## Round 0.1 · original submission · Minor Revisions

Dear Salvador,

Many thanks for your manuscript, and we would like to accept your manuscript, pending review of a few minor points addressed by the reviewers. Rather than accept as is, I hope that the minor points can be addressed, just to clarify a few overlooked details, clarifications and perhaps points of future work. Non of these points should require additional laboratory work. Please specifically address the following points from reviewers, in addition to my own:

Myself:
Please provide all oligonucleotide sequences throughout, both for real -time PCR and cloning strategies.

Reviewer 2:
1) pph/polyhedrin abbreviation
2) Points of GFP reporter protein quality/vs quantity - addressing that this is a consideration would suffice.
3) As a minimum detail the real time PCR protocol for baculovirus quantitation
4) & 5) address briefly

Reviewer 3:
It would be useful (in the absence of relevant data) to state that other recombinant proteins in addition to GFP, should be trialled for expression to gain a full appreciation for the benefit to the community of this new promoter, or such like.

Many thanks!

Reviewer 1 ·

Basic reporting

no comments

Experimental design

no comments

Validity of the findings

no comments

Additional comments

Very good paper, experimental design and results as well as discussion are solid and clearly presented; high scientific relevance, manuscript definitely meets the scope of the journal,

·

Basic reporting

No comments.

Experimental design

No comments.

Validity of the findings

No comments.

Additional comments

The manuscript “A novel baculovirus-derived promoter with high activity in the Baculovirus Expression System” by María Martínez-Solís and colleagues describes the use of the SeMNPV polyhedron envelope protein (PEP) promoter as a potential alternative to the polyhedrin (pph) promoter). The authors present a detailed comparison of the “new” promoter and the strength of the promoter by examining the level of GFP expression in cells, as determined by fluorescence intensity. The study can be published as is; however, it may be worthwhile reflecting on the following points and consider some minor revisions:

1) I am familiar with polh as the abbreviation for polyhedrin and not pph. Is there a reason for using this abbreviation? Are the authors familiar with other works using their method of abbreviation?

2) How suitable is GFP fluorescence as an indicator of the level of GFP protein production? No secondary evidence is provided (SDS/Western) to show that the quantity/level of protein is indeed improved (to be fair, we’ve relied in the past on the fluorescent intensity as well, though one should be aware that it is not the best measure). With promoters, and particularly when talking about the strength of promoters, one needs to recognize that there is a quality vs quantity issue that may arise. In other words is the quality of the protein the same when the expression is driven by pph promoter compared to with the PEP promoter? In this case fluorescence is a measure of quality, but can we infer the quantity from the quality of the protein produced? Also, what quantities of proteins does this represent and how does it compare to other work (in terms of protein per cell?).

3) Groups I have been involved with have done extensive work on quantifying baculoviruses (Shen et al., 2002; George et al., 2012; Transfiguracion et al., 2011), looking at co-infection and co-expression strategies (Sokolenko et al., 2012) and comparing promoter strength (George et al., 2015; George and Aucoin, 2015). One of our primary concerns when doing such studies is that we have equivalent numbers of baculoviruses in each study because of the influence of the number of baculoviruses on the outcome. In this work, little information is provided on the quantification of the baculovirus other than stating that it is relying on a PCR based method. The paper would be strengthened by detailing what primers/sequences were used for quantification and what was used to produce the standard curve. Furthermore, MOIs based on PCR assays are actually much lower as the infectious to total genome numbers are always different by at least one order of magnitude. It would have been nice to show the viability and cell density profiles which are indicative of successful infections. In the images shown in Figure 3 for pSeS, there seems to be a poorer infection (the cells look healthier and more crowded than in the other images). This might not be the case, but viability and density profiles for the cultures could confirm and reinforce that these differences are not based on the baculoviruses but because of the promoters.

4) It would also be good if the authors could specify the time-post-infection when “[e]xpression levels of the SeMNPV genes were monitored by mapping of the viral reads on the transcriptome of S. exigua infected larvae (Pascual et al., 2012)” so that the readers don’t have to refer back to Pascual et al. (2012).

5) The authors repeat a statement that they attribute to Min and Bishop (1991) that states that the polyhedrin gene has an important function for mRNA transcription and translation efficiencies. Does it make sense that it affects translation? Why? It would be good to justify this statement or justify its use.

Reviewer 3 ·

Basic reporting

The manuscript is well written and easy to follow. The figures are presented simply and are readily interpreted. Raw data has been supplied for the figures.There is good referencing to previously published data, with attention paid to quite old papers, which attribute credit to early work - this is not always the case with some studies.

Experimental design

The primary research is within the scope of the journal. The research question is well defined and the authors present their work well, with logical progression from basic analysis of SeNPV gene expression to construction and use of the vectors for GFP production. The investigation has been performed to a high technical standard. The detail provided in the Methods section would make it possible to reproduce the work, although there is perhaps some ambiguity in the nature of the sequence between the various promoters and the GFP reporter.

Validity of the findings

The data presented in the figures is robust, unambiguous and statistically sound. The system devised with a modified gene promoter has been tested in three cell lines with broadly similar results.

Additional comments

To continue from my comments above, the claims for the application of this new promoter do need more supporting evidence. This should not hold up publication of this work, but I hope that further studies are in progress. The claims that the pSel120 promoter might be up to three times better than the AcMNPV polh promoter alone is a little difficult to believe – although well supported by the GFP data. What is lacking is trials with other recombinant proteins. Would such an improvement be seen if the modified promoter was used to express native polyhedrin? The level of this protein, particularly in Hi5 cells, is so high it is hard to see how much more could be made. What about a difficult to expression glycoprotein? The barriers to expression are often not high transcription, but the translation and post translational processing of the protein. Too much protein can sometimes be a bad thing as it becomes insoluble and difficult to purify.

---

## Round 0.2 · accepted · Accept

Hi there,

Just a couple of minor (missed, I think) points - please replace pph to polh in Figures 1 and 3. The website figure legends also state pph not polh, but I think this is just the website - the updated manuscript looks fine for this. I think the other figures have already had pph substituted to polh already. These issues should be addressed in production

Many congratulations! Thanks, Chris